# Prognosis of Subcutaneous Mastectomy for Special Types of Breast Cancer

**DOI:** 10.3390/medicina58010112

**Published:** 2022-01-12

**Authors:** Tsuyoshi Nakagawa, Goshi Oda, Hiroki Mori, Noriko Uemura, Iichiro Onishi, Noriko Sagawa, Tomoyuki Fujioka, Mio Mori, Kazunori Kubota, Toshiaki Ishikawa, Kentaro Okamoto, Hiroyuki Uetake

**Affiliations:** 1Department of Breast Surgery, Tokyo Medical and Dental University, Tokyo 1138519, Japan; oda.srg2@tmd.ac.jp (G.O.); sagawa.srg2@tmd.ac.jp (N.S.); 2Department of Plastic Surgery, Tokyo Medical and Dental University, Tokyo 1138519, Japan; moriplas@tmd.ac.jp (H.M.); noriplas@tmd.ac.jp (N.U.); 3Department of Pathology, Tokyo Medical and Dental University, Tokyo 1138519, Japan; iichpth2@tmd.ac.jp; 4Department of Radiology, Tokyo Medical and Dental University, Tokyo 1138519, Japan; fjokmrad@tmd.ac.jp (T.F.); m_mori_116@yahoo.co.jp (M.M.); kbtmrad@tmd.ac.jp (K.K.); 5Department of Specialized Surgeries, Graduated School, Tokyo Medical and Dental University, Tokyo 1138519, Japan; ishi.srg2@tmd.ac.jp (T.I.); okasrg2@tmd.ac.jp (K.O.); h-uetake.srg2@tmd.ac.jp (H.U.)

**Keywords:** breast cancer, special type, subcutaneous mastectomy, lobular carcinoma, mucinous carcinoma

## Abstract

*Background and objectives:* In the treatment of the special type of breast cancer (STBC), the choice of chemotherapeutic agents is often based on the characteristic features of the histological type. On the other hand, the surgical strategy is usually determined by the tumor size and presence of lymph node metastasis, and the indication for immediate reconstruction is rarely discussed based on the histological type. The prognoses of STBC and invasive ductal carcinoma of the breast (IDC) patients who underwent subcutaneous mastectomy (SCM) with immediate reconstruction at our institution were compared. *Materials and Methods:* A total of 254 patients with SCM with immediate reconstruction from 1998 to 2018 were included; their tumor diameter or induration was less than 25 mm, and it was not in close proximity to the skin. Preoperative chemotherapy and non-invasive cancer cases were excluded. *Results:* The number of patients was 166 for skin-sparing mastectomy (SSM) and 88 for nipple-sparing mastectomy (NSM). The reconstructive techniques were deep inferior epigastric artery perforator flap (DIEP) reconstruction in 43 cases, latissimus dorsi flap reconstruction (LDflap) in 63 cases, tissue expander (TE) in 117 cases, and transverse rectus abdominis myocutaneous flap/vertical rectus abdominis myocutaneous flap (TRAM/VRAM) reconstruction in 31 cases. The histological types of breast cancer were 211 IDC and 43 STBC; 17 were mucinous carcinoma (MUC), 17 were invasive lobular carcinoma (ILC), 6 were apocrine carcinoma, 1 was tubular carcinoma, and 2 were invasive micropapillary carcinoma. There was no difference in local recurrence or disease-free survival (LRFS, DFS) between IDC and STBC, and overall survival (OS) was significantly longer in STBC. OS was better in the STBC group because SCM with immediate reconstruction was performed for STBC, which is a histological type with a relatively good prognosis. Highly malignant histological types, such as squamous cell carcinoma or metaplastic carcinoma, were totally absent in this study. *Conclusions:* The indications for SCM with immediate reconstruction for relatively common STBCs such as MUC and ILC can be the same as for IDC.

## 1. Introduction

Subcutaneous mastectomy (SCM) is a method of breast cancer surgery performed for breast reconstruction and is classified as skin-sparing mastectomy (SSM) or nipple-sparing mastectomy (NSM) [1,2,3,4]. In both cases, the skin just above the tumor is preserved, which makes it difficult to perform for advanced breast cancer. When performing breast cancer surgery, the surgical method is mainly determined by the preoperative tumor diameter and the presence of lymph node metastasis. Special type of breast cancer (STBC) refers to breast cancer other than ductal carcinoma, and there are various histological types [5]. The most common types of STBC are mucinous carcinoma (MUC) and invasive lobular carcinoma (ILC) [5]. It is difficult to diagnose the extent of ILC spread preoperatively, and when partial resections are performed, additional resection is often required after partial resection for ILC [6,7]. Highly malignant histological types of breast cancer, such as squamous cell carcinoma and metaplastic carcinoma, are often not early-stage breast cancer at the time of diagnosis, and breast reconstruction may not be indicated in the primary setting [8,9,10]. The indications and prognosis of SCM with immediate reconstruction for STBC have not been widely discussed. In this study, the prognoses of IDC and STBC cases treated with SCM were compared, and the oncological safety of the procedure was investigated retrospectively. These were evaluated by DFS, OS, and LRFS. MUC and ILC, which are more frequent in STBC, were also compared.

## 2. Methods

### 2.1. Patient Population and Surgical Treatment

A total of 363 patients underwent SCM (231 SSM and 132 NSM) with immediate reconstruction at Tokyo Medical and Dental University from 1998 to 2018. Invasive breast cancer with an invasive component of 1 mm or more was included in the study, but cases with preoperative chemotherapy and non-invasive ductal carcinoma were excluded. Under the exclusion criteria, 254 cases (166 SSM and 88 NSM) were included in the study. SCM was indicated for patients with tumors less than 25 mm in diameter and not in close proximity to the skin or pectoral muscles on preoperative imaging. NSM was indicated for patients with a nipple–tumor distance of at least 1 cm. These indication criteria were defined in 1998. There was no nipple areola necrosis in NSM, and no additional nipple areola excision was performed for positive nipple side margins. Reconstruction techniques included autograft reconstruction or artificial reconstruction, and the patient chose the reconstruction method. Deep inferior epigastric artery perforator flap (DIEP) reconstruction was started in 2008, and SSM was performed in all DIEP cases. This is because microvascular anastomosis is necessary. Sentinel lymph node biopsy (SNB) was introduced in 2004. All cases before that time were treated with axillary lymph node dissection.

### 2.2. Pathological Analysis

Histopathological examination was performed according to the International Union Against Cancer Tumor-Node-Metastasis classification criteria [11]. Blood vessel invasion and lymphatic vessel invasion were also evaluated. The nuclear grade was defined according to the Japan National Surgical Adjuvant Study of Breast Cancer (NSAS-BC) protocol [12]. Biological features, including estrogen receptor (ER) and c-erbB-2 (HER2), were evaluated by immunohistochemistry.

### 2.3. Follow-Up

All patients were followed-up by physical examination every 6 months and mammography with breast ultrasonography annually. Chest and abdominal computed tomography, liver ultrasonography, bone scan, and other investigations were performed in symptomatic cases or whenever clinically indicated. Adjuvant chemotherapy and hormonal therapy were performed according to the contemporary recommendations. Post-mastectomy radiotherapy (PMRT) was carried out according to the pathological findings of each case. Although local recurrence usually includes recurrence in the regional lymph nodes, only local recurrence in the anterior chest wall was considered in this study.

### 2.4. Statistical Analysis

The χ^2^ test and Fisher’s exact test were used to examine the relationships between factors. Median follow-up time was calculated as the median observation time of all patients. Local recurrence-free survival (LRFS) and disease-free survival (DFS) were measured from the date of surgery to the date of first local or distant recurrence. Overall survival (OS) was measured from the date of surgery to the date of death. The actuarial rates of LRFS, DFS, and OS were calculated according to the Kaplan–Meier method. A *p*-value < 0.05 was considered significant. These statistical analyses of clinicopathological and biological factors were carried out using EZR (Saitama Medical Center, Jichi Medical University, Saitama, Japan), which is a graphical user interface for R (The R Foundation for Statistical Computing, Vienna, Austria) [13]. More precisely, it is a modified version of R commander designed to add statistical functions frequently used in biostatistics.

## 3. Results

### 3.1. Comparison of Prognosis between IDC and STBC

The patients’ background characteristics are listed in Table 1. The mean patient age was 46 years (range 26–72 years). The patients had a median follow-up time of 135 months (range 4–280 months). There were 211 cases of IDC and 43 cases of STBC. The special types included 17 cases of MUC, 17 cases of ILC, 6 cases of apocrine carcinoma, 1 case of tubular carcinoma, and 2 cases of invasive micropapillary carcinoma. The nuclear grade of the STBC group was significantly lower, and the tumor diameter tended to be larger.

There were no differences in DFS and LRFS (Figure 1A,B). There were no deaths in the STBC group, and there was a significant difference in OS (Figure 1C).

### 3.2. Comparison of Prognosis between MUC and ILC

The background characteristics of MUC and ILC patients are shown in Table 2. Ten cases of MUC and five cases of ILC had an IDC component. The tumor size of the ILC group was significantly larger. ILC had five pT3 cases, two of which were treated with PMRT.

There were many cases in which NSM was performed in the MUC group. There were no distant recurrences, deaths, or local recurrences of ILC. There were no significant differences between MUC and ILC in DFS, OS, and LRFS (Figure 2A–C).

## 4. Discussion

In general, SCM with immediate reconstruction is indicated for relatively early-stage breast cancer [1,2,3,4]. Therefore, the indication for SCM often depends on tumor size and the presence of lymph node metastasis. Of course, if the patient has advanced cancer, the appropriateness of SCM after preoperative chemotherapy will be considered [14,15].

There are few cases where the indication for SCM is considered based on the histological type of breast cancer. The appropriateness of SCM for STBC has rarely been discussed. In the present study, the prognosis of patients who underwent SCM for STBC was reviewed. However, there is no similarity in the biological behavior of breast cancers that are not IDC [5]. STBC includes squamous cell carcinoma and metaplastic carcinoma, which have very poor prognoses, as well as tubular carcinoma and MUC, which have relatively good prognoses [8,9,10,16,17]. As expected, squamous cell carcinoma or metaplastic carcinoma were totally absent in this study. All of the histological types of the SCMs performed at our institution for STBC were histological types that are considered to have a relatively good prognosis. This may be the most important reason for the better OS of STBC.

In STBC, ILC is the most common histological type that is considered for surgical treatment. ILC accounts for about 5% of primary breast cancers and has increased in recent years. The frequency of hormone receptor expression is high, endocrine therapy is effective, and the prognosis is relatively good [18,19]. On the other hand, the appropriateness of partial mastectomy has often been discussed, and various imaging studies have reported difficulties in diagnosing spread within the breast [6,7]. Additional resection is often required after partial resection for ILC. However, there were no cases of local recurrence in the present study because the whole mammary gland is usually excised when immediate reconstruction is performed. However, ILC cases in which SCM was performed were significantly more likely to have larger tumor diameters, and pT3 cases were relatively common. It may make sense to perform SCM with immediate reconstruction for ILC if the patient is not concerned about partial resection.

MUC accounts for about 3% of primary breast cancers, and pure MUC has a relatively good prognosis [20]. The prognosis of the mixed type of MUC, which are MUCs with an IDC component, is similar to that of normal IDC [21]. In the present study, there were significantly more cases in which NSM was indicated for MUC. This may be because the preoperative imaging often shows that there is little intraductal extension and the tumor is localized. Local recurrence occurred in two MUC cases: one case was a mixed type, and local recurrence occurred through the needle tract at the time of core needle biopsy. In the cases of NSM performed for MUC, fortunately, no local recurrence occurred on the nipple areola complex. In particular, when NSM is performed for Asian patients with small breasts who have mucinous carcinoma with a large vertical diameter, it is necessary to be very careful even if an experienced surgeon is performing NSM, because the field of view of NSM is poor, and cutting into the tumor itself may cause dissemination. Thus, MUC with a large tumor size may increase the possibility of local dissemination, and a total mastectomy should be performed rather than SCM.

This study is limited by the small number of cases at a single institution. A multi-institutional study will be necessary to accumulate cases of SCM for various STBCs, including histological types with poor prognoses. In addition, this study has more limitations, such as the retrospective nature of the study and the included relatively new cases with different follow-up periods.

## 5. Conclusions

The cases of SCM for STBC were limited to histological types with a relatively good prognosis. The indications for SCM with immediate reconstruction for relatively common STBCs such as MUC and ILC can be the same as for IDC.

## Figures and Tables

**Figure 1 medicina-58-00112-f001:**
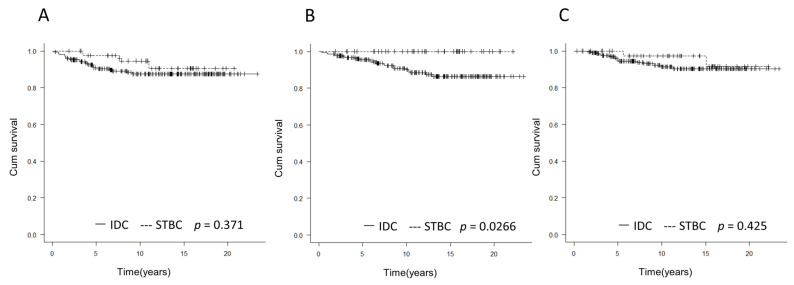
Comparison of prognosis between IDC and STBC. (**A**) DFS: There was no significant difference in DFS between the IDC and STBC groups. (**B**) OS: There were no deaths in the STBC group, and OS was significantly longer in the STBC group (*p* = 0.0266). (**C**) LRFS: There was no significant difference in LRFS between the IDC and STBC groups.

**Figure 2 medicina-58-00112-f002:**
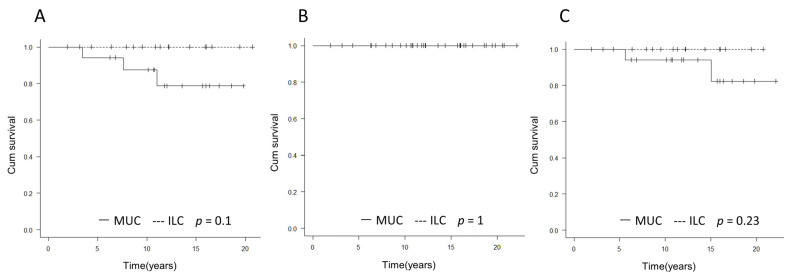
Comparison of prognosis between MUC and ILC. (**A**) DFS, (**B**) OS, (**C**) LRFS. There were no statistically significant differences in DFS, OS, or LRFS between the MUC and ILC groups.

**Table 1 medicina-58-00112-t001:** Background characteristics of all patients.

Characteristics		Total (254)	IDC (211)	STBC (43)	*p*
Age (Mean)		26–72 (46)	26–72 (45)	31–64 (48)	
Type of subcutaneous mastectomy	Skin sparing mastectomy	166	145	21	1
	Nipple sparing mastectomy	88	75	13	
Operation on axillary lymph nodes	Sentinel Lymph node biopsy	129	113	16	0.493
	Sentinel Lymph node biopsy → Axillary lymph node dissection	45	37	8	
	Axillary lymph node dissection	80	70	10	
Reconstruction method	TRAM/VRAM	31	26	5	0.623
	DIEP	43	38	5	
	LD	63	54	9	
	TE	117	102	15	
Post mastectomy irradiation	No	247	215	32	0.338
	Yes	7	5	2	
Histological type	Invasive ductal carcinoma	211	211	-	-
	Special type	43	-	43	
	Mucinous carcinoma	17	-	17	
	Invasive lobular carcinoma	17	-	17	
	Apocrine carcinoma	6	-	6	
	Tubular carcinoma	1	-	1	
	Invasive micropapillary carcinoma	2	-	2	
Tumor size	1	150	137	13	0.0651
	2	94	78	16	
	3	10	5	5	
Lymph node metastasis	No	179	155	24	0.856
	Yes	75	65	10	
Nuclear grade	1	143	113	30	0.0004
	2	59	55	4	
	3	52	52	0	
Estrogen receptor	Positive	223	192	31	0.441
	Negative	31	28	3	
HER2	Positive	20	19	1	1
	Negative	234	201	33	
Lymphatic vessel invasion	Positive	36	32	4	0.811
	Negative	218	188	30	
Blood vessel invasion	Positive	17	16	1	0.32
	Negative	237	204	33	
Adjuvant chemotherapy	No	191	164	27	1
	Yes	63	56	7	
Distant Metastasis	No	228	197	31	0.586
	Yes	26	23	3	
Survival	Alive	233	199	34	0.03
	Dead	21	21	0	
Local Recurrence	No	237	205	32	0.745
	Yes	17	15	2	

**Table 2 medicina-58-00112-t002:** Background characteristics of patients with MUC and ILC.

Characteristics		Total (34)	Mucinous Carcinoma (17)	Invasive Lobular Carcinoma (17)	*p*
Age (Mean)		31–64 (48)	31–62 (45)	38–64 (50)	-
Type of subcutaneous mastectomy	Skin sparing mastectomy	21	7	14	0.0324
	Nipple sparing mastectomy	13	10	3	
Operation on axillary lymph nodes	Sentinel Lymph node biopsy	16	8	8	0.18
	Sentinel Lymph node biopsy → Axillary lymph node dissection	8	2	6	
	Axillary lymph node dissection	10	7	3	
Reconstruction method	TRAM/VRAM	5	2	3	0.223
	DIEP	5	1	4	
	LD	9	7	2	
	TE	15	7	8	
Post mastectomy irradiation	No	32	17	15	0.485
	Yes	2	0	2	
Pure/Mixed type	Pure	19	7	12	0.166
	Mixed	15	10	5	
Tumor size	1	13	6	7	0.0262
	2	16	11	5	
	3	5	0	5	
Lymph node metastasis	No	24	14	10	0.259
	Yes	10	3	7	
Nuclear grade	1	30	15	15	1
	2	4	2	2	
	3	0	0	0	
Estrogen receptor	Positive	31	14	17	0.227
	Negative	3	3	0	
HER2	Positive	1	1	0	1
	Negative	33	16	17	
Lymphatic vessel invasion	Positive	4	2	2	1
	Negative	30	15	15	
Blood vessel invasion	Positive	1	0	1	1
	Negative	33	17	16	
Adjuvant chemotherapy	No	27	16	11	0.0854
	Yes	7	1	6	
Distant Metastasis	No	31	14	17	0.227
	Yes	3	3	0	
Survival	Alive	34	17	17	1
	Dead	0	0	0	
Local Recurrence	No	32	15	17	0.485
	Yes	2	2	0	

## Data Availability

The datasets analyzed in the present study are available from the corresponding author upon reasonable request.

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
