# Peer review of "Prognosis of Subcutaneous Mastectomy for Special Types of Breast Cancer"

_medicina, 2022, doi:10.3390/medicina58010112_

Round 1
Reviewer 1 Report
I congratulate the authors for their interesting work. Analyzing the prognosis of subcutaneous mastectomy for (some of) special types of breast cancer (STBC) and invasive ductal carcinoma (IDC) could be interesting for offering immediate breast reconstruction to these patients.
Nevertheless, a few things can be improved or modified in order to clarify some aspects.
- I recommend that the objective and the primary endpoints for this study should be clarified. - lines 61-62
- Is this a retrospective study? - should be clearly stated.
- How were these patients selected - all the database was searched for? only the patients of the senior author? what exactly were the inclusion criteria? or the exclusion criteria? - lines 65-77. Were these criteria defined in 1998 or in 2018/2019?
- how was the follow-up period defined - patients from 2018 were followed up for 1-2 years and the ones from 1998 for 20 years? the significance for these intervals is not the same
- Why "There were no cases of highly 39 malignant squamous cell carcinoma or metaplastic carcinoma"? were these cases excluded from the study or totally absent?
- Clarify please thisd statement "As expected, there were no cases of squamous cell 154 carcinoma or metaplastic carcinoma with SCM in the present study." - line 154-155. This could be o confounding factor (there were no SCM for this type of cancer or were these cases not included because they did not benefit from SCM).
- Please clarify - what is the significance of the methods described for reconstruction - no discussion is done regarding these data?? Are they only independent variables, like age? This should be clarified and there is no relevance for these data in the abstract section.
- "This study is limited by the small number of cases at a single institution. A multi-institutional study will be necessary to accumulate cases of SCM for various STBCs, including histological types with poor prognoses." I believe that there are also other limitations - like the retrospective nature, the different periods for follow-up. Also, the proposed multi-institutional study should be prospective in design in order to be relevant; blinding and double-armed could offer more valid data but can raise ethical issues.
- minor spelling and editing should be done - lines 25-26 - "in 1998-2018"
- References include a limited number of titles, the majority more than 5 years old.
Author Response
We wish to express our appreciation to the Reviewer for his or her insightful comments, which have significantly helped us to improve the paper.
1.I recommend that the objective and the primary endpoints for this study should be clarified. - lines 61-62
Response: I added “These were evaluated by DFS, OS, and LRFS.” in line 63.
2.Is this a retrospective study? - should be clearly stated.
Response: I added “retrospectively” in line 62-63.
3.How were these patients selected - all the database was searched for? only the patients of the senior author? what exactly were the inclusion criteria? or the exclusion criteria? - lines 65-77. Were these criteria defined in 1998 or in 2018/2019?
Response: Following your suggestion, I have corrected " Patient population and surgical treatment" (lines 67-81).
- how was the follow-up period defined - patients from 2018 were followed up for 1-2 years and the ones from 1998 for 20 years? the significance for these intervals is not the same
Response: As you pointed out, the observation duration is different. I think we should originally include patients who have been followed for at least 5 years. Since the number of cases of special type breast cancer is not large, we have included cases with an observation period of about 2 years. I have also added an explanation to the "Limitation" section at the end of the discussion.
5.Why "There were no cases of highly 39 malignant squamous cell carcinoma or metaplastic carcinoma"? were these cases excluded from the study or totally absent?
Response: I added “Highly malignant histological types, such as squamous cell carcinoma or metaplastic carcinoma, were totally absent in this study” in line 39. In this study, squamous cell carcinoma and metaplastic carcinoma were totally absent in this study.
6.Clarify please thisd statement "As expected, there were no cases of squamous cell 154 carcinoma or metaplastic carcinoma with SCM in the present study." - line 154-155. This could be o confounding factor (there were no SCM for this type of cancer or were these cases not included because they did not benefit from SCM).
Response: I changed the wording in lines 158-159. Squamous cell carcinoma or metaplastic carcinoma, were totally absent in this study.
7.Please clarify - what is the significance of the methods described for reconstruction - no discussion is done regarding these data?? Are they only independent variables, like age? This should be clarified and there is no relevance for these data in the abstract section.
Response: In the text and in the tables, I intended to show that there is no bias in the reconstructive methods. So, we consider it as one of the background factors. Of course, we believe that there is no difference in prognosis depending on the reconstructive technique. Would it be better to remove the reconstructive technique in the summary?
8."This study is limited by the small number of cases at a single institution. A multi-institutional study will be necessary to accumulate cases of SCM for various STBCs, including histological types with poor prognoses." I believe that there are also other limitations - like the retrospective nature, the different periods for follow-up. Also, the proposed multi-institutional study should be prospective in design in order to be relevant; blinding and double-armed could offer more valid data but can raise ethical issues.
Response: I've added a sentence to the limitation (lines 191-193).
9.minor spelling and editing should be done - lines 25-26 - "in 1998-2018"
Response: I made a mistake here. It has been corrected to "2018”. Thank you for letting me know.
10.References include a limited number of titles, the majority more than 5 years old.
Response: Thank you for pointing this out. It is true that the cited references are relatively old. However, I could not find any newer literature on special types of breast cancer, so I cited the most recent one among them.
Reviewer 2 Report
Quality of life of women after surgical treatment because of breast cancer in a high level depends on the effect of reproducer operations. Because of that the shown subject is very actual. Study involved 254 cases of SCM, which underwent immediate breast reconstruction, but there is a discrepancy between a period of doing these operations: 1998-2019 (line 27)/1998-2018 (line 67). In a shown bibliography there is not many items published in recent years. I propose to increase a schedule using new items and use this bibliography in enlargement of the discussion.
Author Response
Response to Reviewer 2 Comments
We wish to express our appreciation to the Reviewer for his or her insightful comments, which have significantly helped us to improve the paper.
Study involved 254 cases of SCM, which underwent immediate breast reconstruction, but there is a discrepancy between a period of doing these operations: 1998-2019 (line 27)/1998-2018 (line 67).
Response: I made a mistake in lines 25-26. It has been corrected to "2018”. Thank you for letting me know.
In a shown bibliography there is not many items published in recent years.
Response: Thank you for pointing this out. It is true that the cited references are relatively old. However, I could not find any newer literature on special types of breast cancer, so I cited the most recent one among them.